Uncovering the potential functions of lymph node metastasis-associated aberrant methylation differentially expressed genes and their association with the immune infiltration and prognosis in bladder urothelial carcinoma

Gao Wenzhi
Zhang Jiafeng
Tian Tai
Fu Zhixin
Bai Liangliang
Yang Yifei
Wu Qiangqiang
Wang Wei
http://orcid.org/0000-0003-3747-7067 Guo Yuexian Guoyuexian@aliyun.com
Department of Urology, The Third Hospital of Hebei Medical University , Shijiazhuang , China
Haraguchi Tokuko
Electronic publication date: 2023 Apr 24
Publication date: 2023
Volume: 11
Electronic Location ID: e15284
Received 2023 Jan 19; Accepted 2023 Apr 3
Copyright: © 2023 Gao et al.
Copyright year: 2023
Copyright holder: Gao et al.
License: This is an open access article distributed under the terms of the Creative Commons Attribution License, which permits unrestricted use, distribution, reproduction and adaptation in any medium and for any purpose provided that it is properly attributed. For attribution, the original author(s), title, publication source (PeerJ) and either DOI or URL of the article must be cited.
License URL: https://creativecommons.org/licenses/by/4.0/

Keywords: Bladder urothelial carcinoma, Differentially expressed genes, Methylation, Risk score, Immune cell infiltration, AKAP7

Funding: The authors received no funding for this work.

==============================
Background

Bladder urothelial carcinoma (BLCA) is a malignant tumor of the urinary system. This study aimed to explore the potential role of lymph node metastasis-associated aberrant methylation differentially expressed genes (DEGs) in BLCA.

Methods

CHAMP and limma packages were used to identify lymph node metastasis-associated aberrant methylation DEGs. Univariate Cox analysis and Lasso analysis were performed to identify the signature genes, and multivariate Cox analysis was used to construct the risk score. Subsequently, the molecular characteristics of the signature genes and the relationship between risk score and prognosis, clinical characteristics and immune cell infiltration were analyzed. The signature gene AKAP7 was selected for functional verification.

Results

A novel risk score model was constructed based on 12 signature genes. The risk score had a good ability to predict overall survival (OS). The nomogram constructed based on age, N stage and risk score had a higher value in predicting the prognosis of patients. It was also found that stromal activation in TIME may inhibit the antitumor effects of immune cells. Functional enrichment analysis revealed that ECM receptor interaction and focal adhesion were two important pathways involved in the regulation of BLCA. Immunohistochemistry showed that AKAP7 may be associated with the occurrence, clinical stages and grades, and lymph node metastasis of BLCA. In vitro cell experiments showed that the migration and invasion ability of EJ cells was significantly inhibited after AKAP7 overexpression, while the migration and invasion ability of T24 cells was significantly promoted after AKAP7 knockdown.

Conclusion

The risk score model based on lymph node metastasis-associated aberrant methylation DEGs has a good ability to predict OS and is an independent prognostic factor for BLCA. It was also found that stromal activation in TIME may inhibit the antitumor effects of immune cells. This implicates aberrant methylation modifications as an important factor contributing to the heterogeneity and complexity of individual tumor microenvironments. Functional enrichment analysis revealed that ECM receptor interaction and focal adhesion were two important pathways involved in the regulation of BLCA, which contributed to the exploration of the pathological mechanism of BLCA. In addition, immunohistochemistry showed that AKAP7 may be associated with the occurrence, progression and lymph node metastasis of BLCA. In vitro cell experiments showed that AKAP7 could also inhibit the migration and invasion of cancer cells.

Introduction

Bladder urothelial carcinoma (BLCA) is highly invasive and has high morbidity and mortality. The presence or absence of lymph node metastasis has a very important influence on the treatment strategy and survival prognosis of patients (Lu et al., 2019). The prognosis of BLCA patients with lymph node metastasis is poor (Abufaraj et al., 2021). Epigenetic modifications, especially DNA methylation, play an important role in the regulation of cancer progression (Makabe et al., 2019; Schulz & Goering, 2016). Abnormal DNA methylation is also common in BLCA. Promoter methylation of cancer-related genes is frequently observed in patients with urothelial carcinoma, and the gene methylation rate of some genes gradually increases with the number of bladder recurrences (Guan et al., 2018). These findings suggest that aberrant methylation genes associated with lymph node metastasis in BLCA may play an important role in regulating the progression and prognosis of patients. However, due to the complexity of the pathological mechanisms, the role of the molecular mechanisms of lymph node metastasis-associated aberrant methylation genes in the progression and prognosis of BLCA is not fully understood. Therefore, we explored the underlying mechanisms of lymph node metastasis-associated aberrant methylation genes in BLCA, hoping to contribute to early diagnosis, treatment and clinical prognosis assessment.

In this study, RNA sequencing data, DNA methylation data and patient clinical information were downloaded from TCGA and GEO databases. Subsequently, a comprehensive analysis of methylation data and RNA sequencing data was performed to investigate the underlying mechanisms of lymph node metastasis-associated aberrant methylation genes in BLCA. Furthermore, a risk model for predicting the prognosis of BLCA was constructed based on lymph node metastasis-associated aberrant methylation genes. In addition, AKAP7 was selected for in vitro validation. Ultimately, we found that risk score constructed based on lymph node metastasis-associated aberrant methylation genes was an independent prognostic factor for BLCA. In addition, AKAP7 may be involved in the occurrence of BLCA and lymph node metastasis, which can inhibit the migration and invasion of cancer cells.

Materials and Methods

Source of datasets

Gene expression data and complete clinical annotations for BLCA were searched from GEO and TCGA databases. Methylation data, gene expression data, clinical information, mutation data and copy number variation (CNV) data in TCGA-BLCA dataset were downloaded from UCSC Xena (https://gdc.xenahubs.net). Then, convert the FPKM value to the transcripts per kilobase million (TPM) value. The GSE13507 dataset was downloaded from GEO database. Subsequently, the data was preprocessed to remove samples without clinical follow-up information, samples with unknown survival time or less than 0 days or and no survival status, and samples without N stage. In addition, the probe was converted to gene symbol. For a probe corresponding to multiple genes, the probe was removed. If multiple probes correspond to one gene, the average value was taken. Totally, 19 paracancerous samples, 236 N0 (non-lymph node metastases) samples, and 127 N1–N3 (lymph node metastases) samples were included in the RNA sequencing data of TCGA-BLCA. Additionally, 21 paracancerous samples, 240 N0 stage samples, and 133 N1–N3 stage samples were included in the methylation data of TCGA-BLCA (only use this data for methylation analysis). A total of 58 paracancerous samples, 149 N0 stage samples, and 15 N1–N3 stage samples were included in the GSE13507 dataset. The specific sample information of the TCGA-BLCA and GSE13507 datasets is shown in Table S1. In this study, the TCGA-BLCA dataset was used the as the discovery cohort, and the GSE13507 dataset was used as the validation cohort.

Methylation and mRNA differential analysis based on TCGA-BLCA data

Firstly, 40 samples of N0 stage and 40 samples of N1–N3 stage were randomly selected from the overall methylation data. By quality control of PCA and correlation clustering heatmap, abnormal data were removed. Finally, 22 N0 stage samples and 24 N1–N3 stage samples were selected for differential methylation analysis (Fig. S1). The CHAMP package was used for differential methylation analysis based on N0 and N1–N3 methylation data. In addition, the limma package used for gene differential expression analysis based on N0 and N1–N3 RNA sequencing data. The screening criteria for differentially methylated sites (DMSs) and differentially expressed genes (DEGs) were false discovery rate (FDR) <0.05. In order to obtain the DEGs regulated by abnormal methylation related to lymph node metastasis of BLCA, the intersection of hypomethylated genes and up-regulated DEGs and the intersection of hypermethylated genes and down-regulated DEGs were selected. Subsequently, GO and KEGG functional enrichment analyses were performed on these aberrant methylation modified DEGs based on the David database. The screening criteria was P < 0.05. Moreover, a protein-protein interaction (PPI) network was also constructed using the STRING database to explore the protein interaction relationship of aberrant methylation modified DEGs.

Construction of risk score model

Univariate Cox regression analysis was performed on aberrant methylation modified DEGs to screen out genes associated with prognosis (P < 0.05). Subsequently, these genes were screened using the Lasso method in the glmnet package to reduce the number of genes for the risk score model. λ cut-off value was 0.03657196. The lambda.min was used to screen out aberrant methylation modified DEGs. Finally, multivariate Cox regression model was used to construct the risk score model related to aberrant methylation modified DEGs. The formula for calculating the risk score is: RiskScore=∑i=1n⁡(Expi∗Coefi) (n, number of prognostic genes; Expi: expression value of gene i, Coefi: regression coefficient of gene i). Aberrant methylation modified DEGs used to construct the risk score were defined as signature genes. Patients were divided into high and low risk groups with the median of risk score as the cut-off point. Kaplan-Meier method was used to analyze the survival of high and low risk groups. Time-dependent receiver operator characteristic (ROC) curve was used to validate the prognostic accuracy of the risk score model.

Molecular characterization of signature genes

Patients were divided into high and low expression groups based on the median expression value of a single signature gene in TCGA-BLCA samples. Kaplan-Meier analysis was used to analyze the effect of signature genes on survival. Subsequently, Wilcoxon test method was used to analyze the differential expression of signature genes between cancer and paracancerous based on TCGA-BLCA data. The differential expression validation of signature genes was performed in the cancer and paracancerous group, and N0 and N1–N3 group of the GSE13507 dataset. At the same time, the Human Protein Atlas (https://www.proteinatlas.org/) database was used to find the expression of signature genes at the protein level of cancer and paracancerous tissues. The mutation status of signature genes was analysed using the “maftools” function. In addition, CNV data of BLCA was analyzed to observe the gain and loss of signature genes.

Prediction of transcription factors (TF) and drugs associated with signature genes

TF subsets were obtained from Cistrome (http://cistrome.org/) and corresponding TF expression values were obtained from the TCGA database. TF associated with signature genes were identified by Pearson correlation analysis (|Pearson correlation coefficient| > 0.4, P < 0.01), and a regulatory network was constructed using Cytoscape. Subsequently, based on the DGIdb database (https://dgidb.org/), we screened out the drugs related to the signature genes. After the gene is input into the DGIdb database, drugs related to the gene can be directly screened without design parameters. It is hoped to contribute to the treatment and management of BLCA.

Risk score and clinical characteristics

Multivariate Cox analysis was used to determine whether the risk score was an independent prognostic factor independent of other clinical factors in BLCA. Nomogram is a powerful tool that has been applied to determine individual risk in the clinical setting by integrating multiple risk factors. We predicted the probability of overall survival (OS) at 1-, 3-, and 5-year by combining factors with independent prognostic significance in multivariate Cox analyses. A calibration curve of the nomogram was also drawn to evaluate the prediction accuracy. Subsequently, the potential net benefit of nomogram was analyzed by decision curve analysis (DCA). In addition, differences of risk score in different clinical subgroups were compared.

Risk score and immune cell infiltration

Gene sets marking 23 immune cell types were obtained from Charoentong et al. (2017). Subsequently, enrichment score calculated by ssGSEA analysis was used to represent the relative abundance of each immune cell infiltrate in each BLCA sample. The ESTIMATE algorithm in the “ESTIMATE” package was used to calculate the stromal score, ESTIMATE score, and tumor purity for each BLCA sample.

Functional enrichment analysis

We also performed gene set variation analysis (GSVA) enrichment analysis using the “GSVA” package. The “c2.cp.kegg.v7.2.symbols” was downloaded from the MSigDB database as reference gene set to run GSVA analysis. The limma package was used to calculate differentially expressed pathways (FDR < 0.05). In addition, the differential expression of genes between high and low risk groups was also analyzed by limma package. Selection criteria for FDR < 0.05 and | log2 (fold change) | (| log2 (FC) |) > 1. Subsequently, GO enrichment analysis of DEGs between high and low risk groups was performed using David database (FDR < 0.05).

QRT-PCR immunohistochemical validation in tissue samples

In total, 18 patients with BLCA were included in this study. The cancer tissue samples and paracancerous tissue samples (normal control group) of BLEA patients were collected for qRT-PCR detection. GAPDH was used as the internal reference gene for qRT-PCR. The relative gene expression was calculated by 2−ΔΔCt method. Meanwhile, immunohistochemistry was used to detect gene expression in different groups of tissues (cancer and paracancerous tissues, graded and staged tissues, non-metastasized and metastatic tissues). Then, the results of immunohistochemically stained sections were examined and scored using a point system (Guo et al., 2021). This study was approved by the Biomedical Research Ethics Committee of Peking University First Hospital (Beijing, China), ethics number 2015(977), and written informed consent was obtained from all patients.

Validation at the cellular level

QRT-PCR was used to detect the expression of AKAP7 in Biu-87, 5637, EJ, T24 and Sv-huc-1 cells. Western blot was performed to detect the expression of AKAP7 in protein level. Then, suitable cells were selected to overexpression or knockdown AKAP7. Subsequently, qRT-PCR and western blot were used to verify the success of knockdown and overexpression of AKAP7 in cells. After successful knockdown and overexpression of AKAP7, Transwell assay was used to detect the effect of AKAP7 expression on cell migration and invasion.

Statistical analysis

All statistical analyses were performed in the R software package (R Core Team, 2019). The Kaplan-Meier method was used to analyze the survival of different groups. Wilcoxon test was used to analyze differences of signature gene expression, immune cell infiltration, stromal score, ESTIMATE score, and tumor purity between groups. In RT-PCR validation, the 2−ΔΔCt method was used to calculate the relative expression of genes.

Results

Identification of aberrant methylation modified DEGs

Compared with N0 samples, 87,589 DMSs were identified in N1–N3 samples. The DMSs contained 14,900 genes (4,608 hypermethylated genes and 13,834 hypomethylated genes). Compared with N0 samples, 213 DEGs were identified in N1–N3 samples, including 146 up-regulated DEGs and 67 down-regulated DEGs. The volcano map of differentially methylated genes and DEGs are shown in Figs. 1A and B. The intersection of hypomethylated genes and up-regulated DEGs and the intersection of hypermethylated genes and down-regulated DEGs were selected. A total of 16 hypermethylated and low expression genes, and 83 hypomethylated and high expression genes were obtained (Figs. 1C and 1D). GO and KEGG analysis found that aberrant methylation modified DEGs were involved in regulating a variety of biological processes (Figs. 1E and 1F), which suggested that they had a complex regulatory mechanism in disease progression. Subsequently, a PPI network of aberrant methylation modified DEGs was constructed (Fig. 1G). The PPI network contains 57 aberrant methylation modified DEGs and 101 edges. FN1 had the most interacting genes (22 interacting genes) in the PPI network. In addition, only the interaction score between FN1 and DCN in PPI network was greater than 0.99, and the interaction score was 0.992.

Figure 1 Identification of aberrant methylation modified DEGs.

(A) The volcano map of differentially methylated genes; (B) the volcano map of DEGs; (C) Venn diagram between low expression genes and hypermethylated genes; (D) venn diagram between high expression genes and hypomethylated genes; (E) GO functional enrichment analysis of aberrant methylation modified DEGs; (F) KEGG functional enrichment analysis of aberrant methylation modified DEGs; (G) PPI network of aberrant methylation modified DEGs.

Construction of risk score model associated with aberrant methylation modified DEGs

The effect of lymph node metastasis on survival was analyzed according to N0 and N1–N3 groups, and the results showed that the survival time of patients with lymph node metastasis was shorter (Fig. 2A). To explore the effect of aberrant methylation modified DEGs on the prognosis of BLCA, we constructed risk model to predict patient survival. Univariate Cox regression analysis screened out 55 aberrant methylation modified DEGs associated with BLCA prognosis. Then, 12 signature genes were further screened by Lasso method (Figs. 2B and 2C). The forest plot of 12 signature genes associated with prognosis is shown in Fig. 2D. Among them, ARHGAP29, EFEMP1, EPN2, LAMA2, SLC1A6 and TMEM109 are hypomethylated and high expression genes, and AKAP7, RPS6KA1, STIM2, SULT1C2, TRABD and ZNRD1 are hypermethylated and low expression genes. Multivariate Cox regression was used to construct a risk score model with 12 signature genes. The risk score formula is as follows: Risk Score = −(AKAP7*0.13) + (ARHGAP29*0.03) + (EFEMP1*0.07) + (EPN2*0.31) + (LAMA2*0.16) − (RPS6KA1*0.13) + (SLC1A6*0.08) − (STIM2*0.20) − (SULT1C2*0.09) + (TMEM109*0.02) − (TRABD*0.06) − (ZNRD1*0.27). Subsequently, the samples were divided into high and low risk groups according to the median risk score (Fig. 2E). It can be seen that the proportion of death samples in the high risk group was higher (Fig. 2F). The expression heatmap of 12 signature genes in high and low risk groups is shown in Fig. 2G. Kaplan-Meier analysis found that overall survival (OS) was significantly lower in the high risk group (Fig. 2H). The same formula was used to calculate the risk score of tumor samples in the GSE13507 dataset, and they were divided into high and low risk groups according to the median risk score (Fig. S2A). It can be seen that in the GSE13507 dataset, the proportion of death samples in the high risk group was also higher (Fig. S2B). The expression heatmap of 12 signature genes in high and low risk groups is shown in Fig. S2C. The Kaplan-Meier analysis found that OS was also significantly lower in the high risk group (Fig. S2D). In addition, the time-dependent ROC curve found that the risk score had a good ability to predict OS with BLCA, and the AUC values of 1-, 3-, and 5-years were 0.692, 0.712 and 0.717, respectively (Fig. 2I). Meanwhile, time-dependent ROC curve analysis was also performed on the GSE13507 dataset (Fig. S2E). The result also showed that risk score had a good ability to predict OS with BLCA.

Figure 2 Construction of prognostic risk score model associated with aberrant methylation modified DEGs in TCGA-BLCA date.

(A) The Kaplan-Meier method was used to analyze the survival of patients in N0 and N1–N3 groups; (B) the trajectory of each independent variable. The horizontal axis represents the log value of the independent variable lambda, and the vertical axis represents the coefficient of the independent variable. (C) Confidence intervals under each lambda; (D) forest map of 12 signature genes; (E) distribution map of risk score; (F) survival status map of patients in high and low risk groups; (G) expression heatmap of 12 signature genes in high and low risk groups; (H) the Kaplan-Meier method was used to analyze the survival of patients in high and low risk groups; (I) time-dependent ROC curves were used to validate the prognostic accuracy of the risk score model.

Molecular characterization of signature genes

Kaplan-Meier analysis found that the OS of patients in the high expression groups of RPS6KA1 and ZNRD1 were significantly higher than that in the low expression groups (Figs. 3A and 3B), while the OS of patients in the high expression groups of TMEM109, LAMA2, EPN2 and EFEMP1 were significantly lower than that in the low expression groups (Figs. 3C–3F). It was not found that the high and low expression of ARHGAP29, STIM2, SULT1C2, TRABD, SLC1A6 and AKAP7 had a significant effect on the survival of patients. Subsequently, the differential expression of 12 signature genes between cancer and paracancerous were analyzed based on the TCGA-BLCA data. Compared with paracancerous tissues, AKAP7, EFEMP1, EPN2 and LAMA2 were lowly expressed in cancer tissues, while RPS6KA1, SLC1A6, TRABD and ZNRD1 were highly expressed in cancer tissues (Fig. 3G). In addition, boxplots of differential expression of 12 signature genes in N0 and N1–N3 groups were also shown (Fig. 3H). The results showed that AKAP7 was low expressed in BLCA cancer samples and also in N1–N3 samples. SLC1A6 was highly expressed in BLCA cancer samples and also in N1–N3 samples. Therefore, it is speculated that AKAP7 and SLC1A6 play an important role in the occurrence of BLCA and lymph node metastasis. Subsequently, The Human Protein Atlas database was used to analyze the protein expression levels of AKAP7, EFEMP1, EPN2, LAMA2, RPS6KA1, SLC1A6, TRABD and ZNRD1 in cancer and paracancerous tissues. The results showed that the protein expression trend was consistent with the mRNA expression trend (Fig. 3I). Mutation analysis of 12 signature genes found that the mutation rate of LAMA2 was relatively high (Fig. 3J). Moreover, all 12 signature genes have different degrees of gain and loss (Fig. 3K).

Figure 3 Molecular characteristics of signature genes.

(A) The Kaplan-Meier method was used to analyze the survival of patients in RPS6KA1 high and low expression groups; (B) the Kaplan-Meier method was used to analyze the survival of patients in the ZNRD1 high and low expression groups; (C) the Kaplan-Meier method was used to analyze the survival of patients in the TMEM109 high and low expression groups; (D) the Kaplan-Meier method was used to analyze the survival of patients in the LAMA2 high and low expression groups; (E) the Kaplan-Meier method was used to analyze the survival of patients in the EPN2 high and low expression groups; (F) the Kaplan-Meier method was used to analyze the survival of patients in the EFEMP1 high and low expression groups; (G) boxplots of differential expression of 12 signature genes in BLCA cancer and paracancerous (control) tissues; (H) boxplots of differential expression of 12 signature genes in the N0 and N1–N3 groups; (I) the protein expression levels of AKAP7, EFEMP1, EPN2, LAMA2, RPS6KA1, SLC1A6, TRABD and ZNRD1 in BLCA cancer and paracancerous (control) tissues; (J) waterfall chart of 12 signature genes mutation; (K) CNV change frequency of 12 signature genes.

Prediction of TF and drugs associated with signature genes

Totally, 125 TF associated with 12 signature genes were identified by correlation analysis (|Pearson correlation coefficient| > 0.4, P < 0.01). Subsequently, a regulatory network was constructed to clearly demonstrate the regulatory relationship between signature genes and TF (Fig. 4A). EFEMP1 had 52 associated TF, SULT1C2 had no associated TF. EFEMP1 had the strongest negative correlation with FOXA1 (−0.51), and LAMA2 had the strongest positive correlation with TCF21 (0.68). In addition, drugs related to 12 signature genes were screened based on DGIdb database. The drugs and signature gene linked by color in the Fig. 4B suggest that they are potentially related. The results showed that only LAMA2 and RPS6KA1 were screened for relevant drugs (Fig. 4B). LAMA2 was associated with one drug OCRIPLASMIN, and RPS6KA1 was associated with five drugs CEP-1347, CHEMBL240954, CHEMBL573107, CI-1040 and HESPERADIN.

Figure 4 Screening of transcription factors associated with signature genes (A) and prediction of drugs associated with signature genes (B).

Risk score and clinical characteristics

Multivariate Cox analysis in TCGA-BLCA and GSE13507 datasets showed that age, N stage and risk score were independent prognostic factors for BLCA (Figs. 5A and 5B). Subsequently, the nomogram was constructed by combining age, N stage and risk score to predict the probability of OS (Fig. 5C). The calibration curve shows that the 1-, 3- and 5-year OS predicted by the nomogram have high accuracy (Fig. 5D). Moreover, compared with the DCA curves of risk score, age and N stages at 1, 3, and 5 years, the nomogram showed a higher contribution to predicting the prognosis of patients (Fig. 5E). Meanwhile, nomogram, calibration curve and DCA curves were also constructed in GSE13507 data set for verification (Figs. S2F–S2H). The risk score was significantly different among the age group, status group, stage group, N stage group, and T stage group (Fig. 5F).

Figure 5 Risk score and clinical characteristics.

(A) Multivariate Cox analysis showed that age, N stage and risk score were independent prognostic factors for BLCA in TCGA-BLCA data; (B) multivariate Cox analysis showed that age, N stage and risk score were independent prognostic factors for BLCA in the GSE13507 dataset; (C) nomogram of age, N stage and risk score; (D) calibration curves for nomogram predicting 1-, 3-, and 5-year OS; (E) DCA curves of nomogram, age, N stage and risk score at 1-, 3- and 5-years; (F) differences of risk score in different clinical subgroups.

Immune correlation analysis

To explore the relationship between the risk score and the tumor immune microenvironment (TIME), we used the ssGSEA algorithm to evaluate 23 immune cell infiltration states in TCGA-BLCA. The results showed that most of the 23 immune cells were more infiltrated in the high risk score group (Fig. 6A). However, BLCA patients in the high risk group did not have a matching survival advantage. Tumors with an immune rejection phenotype also had large numbers of immune cells that were located in the stroma surrounding tumor cells rather than penetrating the tumor. Activation of the stromal in TIME is considered to be T-cell inhibition (Chen & Mellman, 2017). Therefore, we hypothesized that stromal activation inhibited the antitumor effects of immune cells in the high risk group. Subsequent analysis showed that epithelial-mesenchymal transition (EMT) was significantly higher in the high risk subtypes, which confirmed our hypothesis (Fig. 6B). At the same time, the ESTIMATE algorithm was used to calculate the stromal score, ESTIMATE score, and tumor purity of BLCA patients in the high and low risk groups. The results showed that in the high risk group, the ESTIMATE score and stromal score were significantly higher than those in the low risk group (Figs. 6C and 6D), but the tumor purity was lower than that in the low risk group (Fig. 6E). This also implies that immune cells were suppressed in the high risk group.

Figure 6 Analysis of TIME and DEGs in high and low risk groups based on TCGA-BLCA date.

(A) Differential analysis of 23 immune cell infiltration in high and low risk groups; (B) difference analysis of EMT infiltration degree in high and low risk groups; (C) difference analysis of ESTIMATE score in high and low risk groups; (D) difference analysis of tromal score in high and low risk groups; (E) difference analysis of tumor purity in high and low risk groups; (F) differential infiltration analysis of CD56bright.natural.killer.cell, Mast.cell, Natural.killer.cell and Plasmacytoid.dendritic.cell in N0 and N1–N3 groups; (G) correlation analysis between 12 signature genes and CD56bright.natural.killer.cell, Mast.cell, Natural.killer.cell and Plasmacytoid.dendritic.cell; (H) GSVA analysis of differences in biological processes between high and low risk score groups; (I) GO functional enrichment analysis of DEGs between high and low risk groups.

The differences of immune cell infiltration between N0 and N1–N3 groups were also analyzed, and only CD56bright.natural.killer.cell, Mast.cell, Natural.killer.cell and Plasmacytoid.dendritic.cell showed significant differences and consistent with the results in the risk score group (Fig. 6F). Subsequently, the correlations between 12 signature genes and CD56bright.natural.killer.cell, Mast.cell, Natural.killer.cell and Plasmacytoid.dendritic.cell were analyzed. The results showed that EFEMP1 were positively correlated with Natural.killer.cell and Plasmacytoid.dendritic.cell (Fig. 6G). According to FDR < 0.05, a total of 92 metabolic pathways were screened by GSVA analysis. Heatmap was constructed by selecting the top 10 active pathways in the high and low risk groups (Fig. 6H). The activity of extracellular matrix (ECM) receptor interaction and focal adhesion pathways were significantly increased in the high risk group. Subsequently, according to FDR < 0.05 and |log2(FC)| > 1, 86 DEGs were identified in the high and low risk group (75 were up-regulated and 11 were down-regulated). GO enrichment analysis found that DEGs in the cellular component (CC) term were mainly distributed in extracellular space, extracellular region, extracellular exosome and extracellular matrix. DEGs in molecular function (MF) were mainly involved in extracellular matrix structural constituent, antigen binding and immunoglobulin receptor binding. Differentially expressed genes in biological process (BP) were mainly involved in immune related biological pathways such as positive regulation of B cell activation, phagocytosis, recognition (Fig. 6I).

Validation of AKAP7

Bioinformatics analysis showed that AKAP7 and SLC1A6 may play an important role in the regulation of BLCA and lymph node metastasis. Moreover, higher expression level of AKAP7 is associated with better patient survival (Stroggilos & Frantzi, 2022). Therefore, AKAP7 was selected for in vitro validation. The cancer and paracancerous tissues of 18 clinical BLCA patients were collected to verify the expression of AKAP7 by qRT-PCR. The forward primer sequence of AKAP7 was 5′-CTGGTAGACATGCCATTTGCT-3′ and the reverse primer sequence was 5′-GCCAGTCGCTCATCTTGTTGTA-3′. The forward primer sequence of the internal reference gene GAPDH was 5′-GAAGGTGAAGGTCGGAGTCAAC-3′, and the reverse primer sequence was 5′-CAGAGTTAAAAGCAGCCCTGGT-3′. The result showed that AKAP7 was lowly expressed in cancer tissues (Fig. 7A). The results of immunohistochemistry also showed that AKAP7 was lowly expressed in cancer tissues (Figs. 7B–7D). In normal and cancer tissues, AKAP7 protein expression staining was mainly located in the cytoplasm, and partly in the nucleus. Normal and cancer tissues have the same staining sites. Due to the different expression level of AKAP7, the degree of staining of cancer and normal tissues was different. Moreover, with the increase of clinical stage and grade, immunohistochemistry showed that the expression level of AKAP7 in cancer tissues decreased (Figs. 7E–7J). These results further suggest that AKAP7 may play an important regulatory role in the occurrence and progression of BLCA. In addition, the expression of AKAP7 was decreased in lymph node metastatic cancer tissues compared with non-lymph node metastatic cancer tissues (Figs. 7K and 7L). The results of score system of immunohistochemical staining sections for different clinical stages, grades and clinical non-lymph node metastases and lymph node metastases cancer tissue samples are shown in Figs. 7M–7O. These results further suggest that AKAP7 may play an important regulatory role in the occurrence, progression and lymph node metastasis of BLCA.

Figure 7 Expression of AKAP7 in clinical cancer and paracancerous samples.

(A) The expression of AKAP7 in clinical cancer and paracancerous tissue samples was verified by qRT-PCR, asterisks (****) represent P < 0.0001; (B and C) the expression of AKAP7 in clinical cancer and paracancerous tissue samples was detected by immunohistochemistry; (D) the results of immunohistochemically stained sections of clinical cancer and paracancerous tissue samples were scored by points system; (E–H) the expression of AKAP7 in cancer tissues of different clinical stages (T1, T2, T3, T4) was detected by immunohistochemistry; (I and J) the expression of AKAP7 in cancer tissues of different clinical grades (low grade, high grade) was detected by immunohistochemistry; (K and L) the expression of AKAP7 in clinical non-lymph node metastases and lymph node metastases cancer tissue samples was detected by immunohistochemistry; (M) the results of immunohistochemically stained sections of cancer tissue samples of different clinical stages were scored by points system; (N) the results of immunohistochemically stained sections of cancer tissue samples of different clinical grades were scored by points system; (O) the results of immunohistochemically stained sections of clinical non-lymph node metastases and lymph node metastases cancer tissue samples were scored by points system. Observation under 100× microscope.

The mRNA expression level of AKAP7 in Biu-87, 5637, EJ, T24 and Sv-huc-1 cells was verified by qRT-PCR, and the results showed that the expression of AKAP7 was the lowest in EJ cells and the highest in T24 cells (Fig. 8A). The protein expression level of AKAP7 in Biu-87, 5637, EJ, T24 and Sv-huc-1 cells was verified by western blot, and the results showed that the expression of AKAP7 was the lowest in EJ cells and relatively high in T24 cells (Fig. 8B). Therefore, EJ and T24 cells were selected for overexpression and knockdown of AKAP7, respectively. Then, the knockdown and overexpression results of AKAP7 were verified by qRT-PCR and western blot (Figs. 8C–8F). Subsequently, the effect of knockdown and overexpression of AKAP7 on cell migration and invasion ability was detected using Transwell assay. After overexpression of AKAP7, Transwell assay showed that AKAP7 could significantly inhibit the migration and invasion ability of EJ cells (Figs. 9A–9D). After knockdown of AKAP7, Transwell assay showed that the migration and invasion ability of T24 cells were significantly promoted (Figs. 9E–9H). These results indicate that AKAP7 plays an important role in the migration and invasion of cancer cells.

Figure 8 Expression of AKAP7 in different cells in vitro.

(A) The RNA level expression of AKAP7 in Sv-huc-1, T24, 5637, Biu-87 and EJ cells was verified by qRT-PCR; (B) the protein level expression of AKAP7 in Sv-huc-1, T24, 5637, Biu-87 and EJ cells was verified by western blot; (C) validation of AKAP7 overexpression results in EJ cells at the RNA level by qRT-PCR; (D) validation of AKAP7 overexpression results in EJ cells at the protein level by western blot; (E) validation of AKAP7 knockdown results in T24 cells at the RNA level by qRT-PCR; (F) validation of AKAP7 knockdown results in T24 cells at the protein level by western blot. Asterisks (***) represent P < 0.001.

Figure 9 The effects of knockdown and overexpression of AKAP7 on cell migration and invasion were detected by Transwell.

(A) Detection of 24 h migration ability of EJ cells after AKAP7 overexpression; (B) 24 h relative migration histogram of EJ cells after AKAP7 overexpression; (C) detection of 24 h invasion ability of EJ cells after AKAP7 overexpression; (D) 24 h relative invasion histogram of EJ cells after AKAP7 overexpression; (E) detection of 24 h migration ability of T24 cells after AKAP7 knockdown; (F) 24 h relative migration histogram of T24 cells after AKAP7 knockdown; (G) detection of 24 h invasion ability of T24 cells after AKAP7 knockdown; (H) 24 h relative invasion histogram of T24 cells after AKAP7 knockdown. Asterisks (***) represent P < 0.001. Observation under 400× microscope.

Discussion

BLCA patients with lymph node metastases have been reported to have a poor prognosis and difficult clinical management and treatment. Therefore, the study of identifying predictive biomarkers will provide new ideas for the diagnosis and management of BLCA, and will help to inform the best choice of systematic treatment. In this study, 99 lymph node metastasis-associated aberrant methylation DEGs were identified based on differential expression analysis. Then, 12 signature genes (ARHGAP29, EFEMP1, EPN2, LAMA2, SLC1A6, TMEM109, AKAP7, RPS6KA1, STIM2, SULT1C2, TRABD and ZNRD1) were identified by univariate Cox analysis and Lasso analysis to construct the risk score model.

Among 12 signature genes, ARHGAP29, EFEMP1, EPN2, LAMA2, SLC1A6 and TMEM109 are hypomethylated and high expression genes, and AKAP7, RPS6KA1, STIM2, SULT1C2, TRABD and ZNRD1 are hypermethylated and low expression genes. High expression of EFEMP1 was significantly associated with high pathological stage, high histological grade, poor prognosis and lymph node metastasis of BLCA (Chen et al., 2021). It has been reported that LAMA2 is not only associated with the prognosis of bladder cancer, but also strongly related to drug sensitivity (Gu et al., 2022; Zhang et al., 2022b). A study showed that SLC1A6 expression levels in BLCA correlated with the proportion of T cells and neutrophils in the TIME and is an unfavorable prognostic factor for patients (27). In this study, EFEMP1, LAMA2 and SLC1A6 were differentially expressed not only in N0 and N1–N3 groups, but also in cancer and paracancerous groups. In addition, low expression of EFEMP1 and LAMA2 were also associated with better survival. This again proves that EFEMP1, LAMA2 and SLC1A6 may play important regulatory roles in BLCA disease. TF analysis of signature genes revealed that EFEMP1 had the strongest negative correlation with FOXA1 (−0.51), and LAMA2 had the strongest positive correlation with TCF21 (0.68). Loss of FOXA1 is associated with high-grade and advanced bladder cancer (DeGraff et al., 2012). A study showed that TCF21 inhibited lymph node metastasis in bladder cancer (Mokkapati, Porten & Narayan, 2020). Therefore, we speculate that EFEMP1 and LAMA2 may be regulated by FOXA1 and TCF21, respectively, in the regulation of BLCA progression and lymph node metastasis. High expression of AKAP7 is associated with better survival in BLCA patients (Stroggilos & Frantzi, 2022). In this study, bioinformatics analysis showed that AKAP7 was low expressed in BLCA cancer samples and also in N1–N3 samples. In addition, AKAP7 was found to inhibit the migration and invasion of cancer cells in vitro. Therefore, we hypothesized that AKAP7 might be a marker in lymph node metastasis of BLCA. So far, there is no relevant report on the study of ARHGAP29, EPN2, TMEM109, RPS6KA1, STIM2, SULT1C2, TRABD and ZNRD1 in bladder cancer. To our knowledge, this is the first study to show that ARHGAP29, EPN2, TMEM109, RPS6KA1, STIM2, SULT1C2, TRABD and ZNRD1 may play a potential regulatory role in lymph node metastasis of BLCA. It is worth noting that the identification of these signature genes provides potential research directions for further research on the pathological mechanism of lymph node metastasis in BLCA.

Based on DGIdb database, six drugs (OCRIPLASMIN, CEP-1347, CHEMBL240954, CHEMBL573107, CI-1040 and HESPERADIN) related to signature genes were screened. CEP-1347, CI-1040 and HESPERADIN have been found to play a role in cancer therapy. In vivo, CEP-1347 was able to induce the differentiation of cancer stem cells into non-cancer stem cells, thereby inhibiting the self-renewal and tumor-initiating capabilities of cancer stem cells. In vivo, CEP-1347 effectively reduced tumor initiating populations and provided a significant survival advantage (Okada et al., 2017). CI-1040 combined with rapamycin and 17-AAG can effectively inhibit the metastatic ability of prostate cancer (Ding et al., 2013). CI-1040 also inhibited the growth of papillary thyroid carcinoma (PTC) cells in vitro and in vivo (Henderson, Ahn & Clayman, 2009). Previous studies have found that HESPERADIN can inhibit the growth of cancer cells (Zhang et al., 2022a; Zhang, Wu & Fu, 2022). So far, no relevant studies on OCRIPLASMIN, CEP-1347, CHEMBL240954, CHEMBL573107, CI-1040 and HESPERADIN in bladder cancer have been found. It is hoped that the identification of these drugs will be helpful for the treatment of BLCA and the inhibition of lymph node metastasis.

A risk model was constructed based on 12 signature genes. The Kaplan-Meier analysis found that OS was significantly lower in the high risk group. Moreover, the time-dependent ROC curve found that the risk score had a good ability to predict OS with BLCA. Multivariate Cox analysis showed that the risk score was an independent prognostic factor for BLCA. We also found that the risk score was significantly different among the age group, status group, stage group, N stage group, and T stage group. Nomogram is a statistical model widely used in cancer prognosis. It can visually display the relevant factors that affect the results of multivariate regression analysis, and can also predict the survival probability through a simple graphical representation, making the prediction more concise and convenient (Liu et al., 2021; Raghav, Hwang & Jácome, 2021). In this study, the nomogram was constructed by combining age, N stage and risk score to predict the probability of OS. Calibration and DCA curves show that the nomogram has high predictive accuracy and value. Therefore, BLCA patients can be divided into different groups based on the risk score model, which may facilitate early detection and personalized treatment. Notably, there are significant differences of TIME between high and low risk groups. In the high risk group, the infiltration degree of most immune cells was higher than that in the low risk group, but there was no corresponding survival advantage. Moreover, subsequent analyses found that EMT was significantly higher in the high risk group. Therefore, we hypothesized that stromal activation inhibited the antitumor effects of immune cells in the high risk group. Previous studies have shown that activation of EMT and TGFbeta-related pathways leads to reduced T-cell trafficking to tumors and their tumor-killing effects (Mariathasan et al., 2018; Tauriello et al., 2018). This suggests that stromal activation in TIME may mediate BLCA immunotherapy. Correlation analysis showed that the risk score was significantly correlated with these four immune cells. This further suggests that this risk score has a good advantage in determining TIME infiltration patterns in individual tumor samples.

GSVA analysis showed that the activity of ECM receptor interaction and focal adhesion pathways were significantly increased in high risk group. The interaction of ECM produced by highly metastatic breast cancer cell MDA-MB-231 can induce EMT in MCF-7 cells, increase the expression of mesenchymal markers, alter cell phenotype, and stimulate cell migration (Brandão-Costa et al., 2020). Moreover, ECM receptor interaction signaling is also associated with cancer development and poor survival (Guo et al., 2022; Yeh et al., 2018). In cancer, focal adhesion kinase is a major driver of invasion and metastasis, and it’s up-regulated is associated with poor prognosis (Tapial Martínez, López Navajas & Lietha, 2020). Focal adhesion signaling plays an important role in cancer growth and metastasis and is also associated with disease resistance (da Silva et al., 2019; Lin et al., 2022; Nakanishi et al., 2018). GO enrichment analysis was performed for DEGs between high and low risk groups based on David database. The results showed that DEGs were also significantly enriched in extracellular matrix of CC term and extracellular matrix structural constituent of MF term. Furthermore, KGGG and GO analyses to understand the biological processes that may be affected by lymph node metastasis-associated aberrant methylation DEGs also suggested that ECM receptor interaction and focal adhesion may play an important role in lymph node metastasis of BLCA. Our data also revealed that DEGs between the high and low risk groups in the BP term were mainly enriched in immune related biological pathways, which indicated the dysregulation of immune regulation in patients. The identification of these immune related biological pathways lays a theoretical foundation for understanding the immune mechanism of BLCA and provides new ideas for promoting cancer immunotherapy in the future.

This study also has some limitations. First of all, the risk score and nomogram obtained in this study are based on public database data, and lack of clinical sample verification. Therefore, a large number of clinical samples should be collected for further validation. Second, the key molecules and signaling pathways identified in this study need to be further validated in vitro to understand their molecular mechanisms in lymph node metastasis of BLCA.

Conclusion

A risk score model was constructed based on 12 lymph node metastasis-associated aberrant methylation DEGs in BLCA. The risk score has a good ability to predict OS and is an independent prognostic factor for BLCA. It was also found that stromal activation in TIME may inhibit the antitumor effects of immune cells. This implicates aberrant methylation modifications as an important factor contributing to the heterogeneity and complexity of individual tumor microenvironments. Functional enrichment analysis revealed that ECM receptor interaction and focal adhesion were two important pathways involved in the regulation of BLCA, which contributed to the exploration of the pathological mechanism of BLCA. In addition, immunohistochemistry showed that AKAP7 may be associated with the occurrence, progression and lymph node metastasis of BLCA. In vitro cell experiments showed that AKAP7 could also inhibit the migration and invasion of cancer cells. In short, the results of this study provide new ideas for promoting individualized cancer immunotherapy in patients, improving patients’ response to clinical treatment, and deepening the understanding of the molecular mechanism of BLCA.

Supplemental Information

Supplemental Information 1 Raw data in Figure 7-9.

Click here for additional data file.

Supplemental Information 2 Screening and quality control of samples in methylation datasets.

A: Principal components analysis (PCA) diagram of 40 N0 methylation samples and 40 N1-N3 methylation samples before quality control; B: Heatmap of 40 N0 methylation samples and 40 N1-N3 methylation samples before quality control; C: PCA diagram of 22 N0 methylation samples and 24 N1-N3 methylation samples after quality control; D: Heatmap of 22 N0 methylation samples and 24 N1-N3 methylation samples after quality control.

Click here for additional data file.

Supplemental Information 3 Validation of risk model in the GSE13507 dataset.

A: Distribution map of risk score in the GSE13507 dataset; B: Survival status map of patients in high and low risk groups; C: Expression heatmap of 12 signature genes in high and low risk groups; D: The Kaplan-Meier method was used to analyze the survival of patients in high and low risk groups; E: Time-dependent ROC curves were used to validate the prognostic accuracy of the risk score model; F: Nomogram of age, N stage and risk score; G: Calibration curves for nomogram predicting 1-, 3-, and 5-year OS; H: DCA curves of nomogram, age, N stage and risk score at 1-, 3- and 5-years.

Click here for additional data file.

Supplemental Information 4 Sample size and clinical characteristics in TCGA-BLCA and GSE13507 datasets.

Click here for additional data file.

Supplemental Information 5 Full-length uncropped blots.

Click here for additional data file.

Supplemental Information 6 Procedure code.

Click here for additional data file.

Supplemental Information 7 Original data source.

Click here for additional data file.

Supplemental Information 8 Immunohistochemistry: cancer and paracancerous.

Click here for additional data file.

Supplemental Information 9 Immunohistochemistry: different clinical stages.

Click here for additional data file.

Supplemental Information 10 Immunohistochemistry: different clinical grades.

Click here for additional data file.

Supplemental Information 11 Immunohistochemistry: metastases and no metastases.

Click here for additional data file.

We thank the Institute of Urology, Peking University for providing the bladder cancer tissues and paraffin blocks for this study, and the participants in the Department of Urology, Third Hospital of Hebei Medical University for their guidance and assistance with this study.

Additional Information and Declarations

Competing Interests

Author Contributions

Human Ethics

Data Availability

The authors declare that they have no competing interests.

Wenzhi Gao conceived and designed the experiments, performed the experiments, analyzed the data, prepared figures and/or tables, authored or reviewed drafts of the article, provision of study materials or patients, and approved the final draft.

Jiafeng Zhang conceived and designed the experiments, analyzed the data, prepared figures and/or tables, authored or reviewed drafts of the article, and approved the final draft.

Tai Tian performed the experiments, analyzed the data, prepared figures and/or tables, authored or reviewed drafts of the article, provision of study materials or patients, and approved the final draft.

Zhixin Fu performed the experiments, analyzed the data, prepared figures and/or tables, authored or reviewed drafts of the article, provision of study materials or patients, and approved the final draft.

Liangliang Bai performed the experiments, analyzed the data, prepared figures and/or tables, authored or reviewed drafts of the article, provision of study materials or patients, and approved the final draft.

Yifei Yang performed the experiments, analyzed the data, prepared figures and/or tables, authored or reviewed drafts of the article, provision of study materials or patients, and approved the final draft.

Qiangqiang Wu performed the experiments, analyzed the data, prepared figures and/or tables, authored or reviewed drafts of the article, provision of study materials or patients, and approved the final draft.

Wei Wang performed the experiments, analyzed the data, prepared figures and/or tables, authored or reviewed drafts of the article, provision of study materials or patients, and approved the final draft.

Yuexian Guo conceived and designed the experiments, analyzed the data, prepared figures and/or tables, authored or reviewed drafts of the article, and approved the final draft.

The following information was supplied relating to ethical approvals (i.e., approving body and any reference numbers):

This study was approved by the Biomedical Research Ethics Committee of Peking University First Hospital (Beijing, China), ethics number 2015(977).

The following information was supplied regarding data availability:

The code and raw data are available in the Supplemental Files. The public gene expression data and complete clinical annotations for BLCA were obtained from GEO (GSE13507) and TCGA (TCGA: BLCA).

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
