# Peer review of "Uncovering the potential functions of lymph node metastasis-associated aberrant methylation differentially expressed genes and their association with the immune infiltration and prognosis in bladder urothelial carcinoma"

_PeerJ, doi:10.7717/peerj.15284_

## Round 0.1 · original submission · Minor Revisions

Reviewers greatly appreciated the value of the paper. But they also pointed out some problems with the manuscript. I hope you will find their suggestions helpful in improving your paper. If you need more time to revise the manuscript, please let me know.

Reviewer 1 ·

Basic reporting

The manuscript by Gao and coworkers titled ‘Uncovering the potential functions of lymph node metastasis-associated aberrant methylation differentially expressed genes and their association with the immune infiltration and prognosis in bladder urothelial carcinoma’ identified genes regulated by methylation changes in bladder urothelial carcinoma. The authors further identified association of the expression of these genes with methylation and prognosis. Authors developed a 12 gene prognosis signature and showed some validation in independent dataset. In addition, they also established the effect of AKAP7 on oncogenic phenotypes of BLCA in invitro experiments.
This study is interesting. I appreciate the authors for conducting a comprehensive bioinformatics analysis and complementing with invitro data. Introduction and discussion were appropriately written. Methods were described adequately clear with few exceptions. Results section requires minor improvement in writing and presentation. This validation of the signature is the weak link in this manuscript and therefore requires refinement.

I ask authors to address the following minor concerns.
1) In abstract, last sentence of results section is largely vague. Please write this sentence clearly. Precisely state what you found. The words occurrence and progression are not substantiated by the results. Please state if the phenotypes increased or decreased with over expression or knockdown.
2) It seems the validation and model calibration were performed on TCGA cohort. This would be overfitting and inflates the model performance. The language in results and figure legends seems inconspicuous on this which is slightly disappointing. Please show complete model validation on GSE13507. Even if the model performance is modest, it is the correct way of showing the data.
3) Why was adjusted p value not used in DAVID analysis? (Line 96)
4) How was LASSO implemented? What package? What is lambda cut off? Which model was selected for 12-gene signature? Is it lambda.min or lamda.1se? These details should be clearly described in the text.
5) Interpretation of figure 1D is not easy. Can these pathways be shown in a conventional manner?
6) Describe staining pattern of immunohistochemistry of AKAP5 in cancer and normal tissues? Are there differences in staining pattern (such as exclusively at invasive front, more nuclear) between tumors and normal.

Experimental design

No comment

Validity of the findings

No comment

Reviewer 2 ·

Basic reporting

This study investigated the genes whose expression is affected by differential methylation in bladder urothelial carcinoma (BLCA). This study discovered a prognosis predictor selected from such genes and seems to have a food accuracy in predicting 1-, 3- and 5-year survival. However, I did not understand if this was overfitting to the discovery set or evaluated on an independent set of samples. Statistical analysis was appropriately conducted though it should improve in its presentation and clarity. Addition of experimental evidence showing AKAP7 as a tumor suppressor increased my excitement. This manuscript requires some minor revisions.
I have the following comments.
1. Validation of prognosis signature in independent dataset was limited to just multivariate regression. It would be nice to see how the signature performs in unseen set of samples and how good is the calibration on such samples.
2. Please add a table showing number of samples and clinical characteristics in both cohorts.
3. Please add color key to figure 1E. Please explain the colors and width of nodes and hubs.
4. Please show median survival values for all survival curves in the manuscript.
5. Please show the higher magnification of IHC pictures (in inset) shown in figure 7. It will help in understanding the nuclear/membrane/cytoplasmic staining.
6. Lines 32 -34: It is not appropriate to write that AKAP7 influences lymph node metastasis. Similarly, it is not appropriate to write that this study provides ideas for personalized immunotherapy. I suggest that authors should write more relevant conclusions to the data.
7. The fonts in figure 6G and figure 6H are too small to read.

Experimental design

Please see basic reporting.

Validity of the findings

Please see basic reporting.

Reviewer 3 ·

Basic reporting

This paper from Dr. Guo’s group reports a prognosis biomarker panel for lymph node metastasis in bladder urothelial carcinoma. This signature predicts prognosis and stratifies patients with differences in several immune cell infiltration. The findings are exciting. The manuscript need improvement in presentation. The major problem with this manuscript is brevity in description of discovery and validation and how did they perform the validation. I suggest authors to rewrite the results and rearrange the figure to ensure that the reader would understand which cohort was used in each figure.
These are the concerns I have. Please address them thoroughly.
1. The methods portion the manuscript shows that two cohorts (TCGA and GSE13507) were used in this manuscript. There is no clarity on GSE69795. Please add details description of this cohort.
2. Was the regression formula derived in TCGA cohort used exactly as it is in the other cohorts? Or did you derive regression equation separately in each cohort?
3. Why were gender, TNM stage etc omitted form multivariate cox analysis on GSE dataset? Keeping the variable would help understand the consistency between cohorts.
4. Add a header for p value column in Figure 5A&B.
5. Why were validations and nomogram performed on the same dataset used for discovery? It will mislead the authors on the accuracy of the signature. Show nomogram, calibration curves and cost-benefit plots for GSE13507 and GSE69795.
6. Show chi square test p values for differences in IHC staining score between different groups in figure 7.
7. Figure 1D and E are not legible. What do the gene names in figure 1D indicate and what is the basis for matching the color of genes and pathways?
8. Elaborate the technical details of finding the drugs associated with genes shown in figure 4b. What are analysis parameters and what is the specificity? Please explain how to interpret figure 4B.

Experimental design

Please refer to basic reporting.

Validity of the findings

Please refer to basic reporting.

---

## Round 0.2 · accepted · Accept

Your responses have addressed all of the reviewer's comments. Therefore, I am happy to accept your manuscript for publication.

Reviewer 2 ·

Basic reporting

My comments were addressed.

Experimental design

My comments were addressed.

Validity of the findings

My comments were addressed.